# Bioinformatics Identification of TUBB as Potential Prognostic Biomarker for Worse Prognosis in ERα-Positive and Better Prognosis in ERα-Negative Breast Cancer

**DOI:** 10.3390/diagnostics12092067

**Published:** 2022-08-26

**Authors:** Rashed Alhammad

**Affiliations:** Department of Pharmacology, Faculty of Medicine, Kuwait University, Safat 13110, Kuwait; rashed.alhammad@ku.edu.kw

**Keywords:** TUBB, bioinformatics, breast cancer, ERα

## Abstract

Tubulin β class I gene (*TUBB*) is highly expressed in various cancers and plays several roles in carcinogenesis. However, the prognostic value of *TUBB* in breast cancer remains to be investigated. GEPIA and Breast Cancer Gene-Expression Miner were used to explore *TUBB* expression in breast cancer patients. Kaplan–Meier Plotter was used to assess the relationship between *TUBB* expression and several prognostic indicators including overall, distant metastasis-free, and relapse-free survival in ERα-positive and ERα-negative breast cancer. The genes that correlate with *TUBB* in ERα-positive and ERα-negative breast cancer were explored and the pathways were investigated using GSCA. The correlation between *TUBB* and several gene markers of immune cells was explored using GEPIA. ERα-positive breast cancer patients with increased *TUBB* showed worse prognosis, possibly through the activation of the TSC/mTOR pathway, whereas ERα-negative breast cancer patients with increased *TUBB* mRNA showed better prognosis. Significant positive correlations were observed between *TUBB* and gene markers of immune cells in ERα-positive breast cancer patients, whereas significant negative correlations were observed in ERα-negative breast cancer patients. The analysis revealed that *TUBB* might be considered as a predictive biomarker for worse prognosis in ERα-positive and better prognosis in ERα-negative breast cancer.

## 1. Introduction

Tubulin β class I gene (*TUBB*) encodes β tubulin protein that belongs to the β-tubulin family, which contains ten isoforms [1]. TUBB forms a dimer with α-tubulin and acts as structural components of microtubules, which are essential for cell division and intracellular signaling and transport [2,3]. During mitosis, TUBB protein is localized in several locations including microtubules, the cytokinetic bridge, and mitotic spindles [4].

The autoregulatory mechanism governs the regulation and stability of *TUBB* mRNA, in which the formation of new *TUBB* mRNA is suppressed and the decay of the existing *TUBB* mRNA is hastened by the increased unpolymerized tubulin pool [5]. In addition, it has been observed that PI3K signaling activity promotes *TUBB* mRNA stabilization [6]. Interestingly, 17β-estradiol has been shown to upregulate *TUBB* mRNA and protein levels [7]. A recent report showed that miR-195 directly targets and inhibits *TUBB* mRNA and protein levels [8]. Additionally, CHEK1 has been shown to regulate TUBB protein levels, in which CHEK1 overexpression was shown to increase TUBB protein levels in lung cancer cell lines [8]. Although the crosstalk between TUBB and Rho/ROCK signaling pathway has not been studied extensively, TUBB has been shown to activate the Rho/ROCK signaling pathway that mediates pancreatic cancer proliferation and metastasis [9].

TUBB has been shown to play various pathological roles, in which elevated levels of TUBB were observed in breast cancer tumors and tumor-adjacent normal breast tissues compared to normal breast tissues [10]. In addition, TUBB is the most highly expressed isoform of β-tubulin in most epithelial tumor cells and associates positively with worse prognosis, metastasis, and confers resistance to microtubule-targeting agents in lung adenocarcinoma [8,11,12]. Moreover, TUBB has been shown to mediate neuroblastoma survival, in which silencing *TUBB* in differentiated neuroblastoma cells caused apoptosis [13]. It has been indicated that TUBB mediates cell cycle, epithelial-mesenchymal transition (EMT), proliferation, metastasis, and invasion in cutaneous melanoma, possibly through the downregulation of tumor suppressor miR-339-3p [14]. It has been shown recently that *TUBB* mRNA expression is upregulated in breast cancer patients resistant to chemotherapy [10,15]. Mutations in *TUBB* have been shown to demonstrate resistance to chemotherapy [16,17], in which the l240I mutation shows specific resistance to vincristine [18]. In contrast, A248V, A185T, and R306C mutations confer resistance to paclitaxel [17]. Additionally, *TUBB* has been shown to be involved in the breast cancer cell cycle [19]. Although high *TUBB* mRNA levels correlate with poor prognosis in renal and liver cancer [20,21], its correlation with breast cancer outcome remains to be investigated.

Given the fact that there is a high demand for new biomarkers to improve individualized treatment regimens and prediction outcomes due to breast cancer heterogeneity and that TUBB is overexpressed in diverse tumors and plays a vital role in carcinogenesis [8,9,13], the aim of this report is to predict the significance of TUBB in ERα-positive versus ERα-negative breast cancer by bioinformatics analyses of databases.

## 2. Materials and Methods

### 2.1. TUBB mRNA Expression in Normal Breast Tissue versus Breast Cancer Tissue

Two different databases were utilized to investigate the mRNA expression of *TUBB* in normal breast tissue versus breast cancer tissue. GEPIA (Gene Expression Profiling Interactive Analysis; https://gepia.cancer-pku.cn) (accessed 2 May 2022), a tool based on TCGA and GTEx data that provides RNA expression data of 9736 tumors and 8587 normal samples was utilized to explore *TUBB* mRNA expression [22]. In the expression analysis, the threshold included expression log_2_ fold change >1.5 between cancer (*n* = 1085) and normal tissues (*n* = 291), *p*-value < 0.05. Breast Cancer Gene-Expression Miner version 4.1, Integrated Center for Oncology (Nantes, France) (http://bcgenex.ico.unicancer.fr/BC-GEM/GEM-Accueil.php?js=1) (accessed 4 May 2022), which is a statistical mining tool of published annotated genomic data was also utilized to assess *TUBB* mRNA levels in breast cancer (*n* = 743) and normal breast tissue (*n* = 92) [23].

### 2.2. Kaplan–Meier Overall Survival Analysis

To assess the prognostic value of *TUBB* gene in ERα-positive and ERα-negative breast cancer patients, the Kaplan–Meier Plotter website (Budapest, Hungary) (http://kmplot.com/analysis/) (accessed 14 May 2022) was utilized [24]. For this purpose, *TUBB* Kaplan–Meier plots for ERα-positive and ERα-negative breast cancer patients were generated. The following parameters were selected: overall survival (OS), distant metastasis-free survival (DMFS), and relapse-free survival (RFS); split patients by: upper quartile, follow up threshold: 240 months, probe set option: all probe sets per gene [25]. Logrank *p*-values < 0.05 for the Kaplan–Meier (KM) plots of all genes was considered statistically significant.

### 2.3. TUBB Correlation with Neoplasm Histologic Grades and Lymph Nodes

The METABRIC breast cancer dataset was downloaded from cBio Cancer Genomics Portal (Massachusetts, United states) (http://cbioportal.org) (accessed 18 May 2022) for 1900 breast cancer patients (Appendix A) [26]. The distribution of the mRNA expression level of *TUBB,* neoplasm histologic grade, and lymph nodes examined positive were measured across samples. *TUBB* mRNA expression levels were considered high if their z-scores were higher or equal to the 75th percentile of the distribution [27].

### 2.4. Identification of Genes That Correlate with TUBB in ERα-Positive and ERα-Negative Breast Cancer and Show Similar KM Plots

Breast Cancer Gene-Expression Miner version 4.1 was utilized to explore the top 50 genes that show significant positive and negative correlation with *TUBB* in ERα-positive and ERα-negative breast cancer (Appendix A) [23]. The patients were split by ERα status and *p*-value < 0.0001 was considered significant.

### 2.5. Association of Genes That Correlate Positively and Negatively with TUBB in ERα-Positive and ERα-Negative Breast Cancer and Show Similar KM Plots with Signaling Pathways

The effect (activation or inhibition) of our genes on pathways related to cancer, including TSC/mTOR, RTK, RAS/MAPK, PI3K/AKT, hormone ER, hormone AR, EMT, DNA damage response, cell cycle, and apoptosis pathways, was assessed using the GSCA web server (Houston, TX, USA) (http://bioinfo.life.hust.edu.cn/GSCA/#/) (accessed 4 May 2022) [28]. Only breast invasive carcinoma patients with available expression data and who had paired samples (paired tumor–normal tissue) were included in the analysis.

### 2.6. Immune Cells Infiltration

Breast Cancer Gene-Expression Miner v4.1 was utilized to assess Pearson’s correlation coefficients between *TUBB* and several gene markers of immune cells, including T cells and B cells [29]. Logrank *p*-values < 0.05 was considered statistically significant. *p*-value < 0.0001 was considered significant.

### 2.7. Identification of Repurposed Drugs

Following the identification of TUBB importance for ERα-positive and ERα-negative breast cancer patient survival, the potential clinical targeting of TUBB was explored via DRUGSURV [30]. The approved drugs that targeted TUBB directly and are currently used in breast cancer treatment were recorded.

### 2.8. Statistical Analysis

PRISM 8 GraphPad software Inc. (San Diego, CA, USA) was used to plot the graphs. The significance of the differences between more than two groups was assessed using the Kruskal–Wallis test, whereas the significance of the difference between two groups was assessed using Mann–Whitney test. The significance of the difference between Kaplan–Meier survival curves was assessed using Log-rank.

## 3. Results

### 3.1. Higher TUBB mRNA in Breast Cancer Patients Compared to Normal Breast Tissue

The mRNA expression of *TUBB* in breast cancer was investigated using several bioinformatic platforms including GEPIA and bc-GenExMiner. In the databases, significantly higher *TUBB* mRNA expression in breast cancer patients was observed, compared to normal breast tissue (Figure 1A,B). Significantly higher *TUBB* mRNA was observed in breast cancer tissue compared to tumor-adjacent and normal tissue (Figure 1A).

### 3.2. Higher TUBB mRNA Expression Correlates with Poor Prognosis in ERα-Positive Patients and Better Prognosis in ERα-Negative Patients

The KM plotter website was utilized to assess the prognostic value of the *TUBB* gene in ERα-positive and ERα-negative breast cancer patients. The analysis revealed that higher *TUBB* mRNA expression significantly correlates with worse prognosis in ERα-positive breast cancer patients, in which it correlates with lower OS (Figure 2A), worse DMFS (Figure 2B), and worse RFS (Figure 2C), whereas *TUBB* mRNA expression significantly correlates with preferable OS (Figure 2D), preferable DMFS (Figure 2E), and preferable RFS (Figure 2F) in ERα-negative breast cancer patients.

### 3.3. TUBB Correlates Differentially with Lymph Nodes and Neoplasmic Histologic Grades in ERα-Positive and ERα-Negative Breast Cancer Patients

To validate our observation that *TUBB* mRNA correlates with worse prognosis in ERα-positive breast cancer patients, METABRIC dataset in the cBio Cancer Genomics Portal was explored. Significant positive correlation was observed between *TUBB* mRNA and lymph nodes examined positive in ERα-positive breast cancer patients (Figure 3A). In contrast, the correlation between *TUBB* mRNA and lymph nodes examined positive was not significant in ERα-negative breast cancer patients (Figure 3B). The analysis showed significant positive correlation between neoplasmic histologic grades and *TUBB* mRNA in ERα-positive breast cancer patients, in which neoplasmic histologic grade 3 showed significantly higher *TUBB* mRNA compared to grades 2 and 1 (Figure 3C). Neoplasmic histologic grade 3 showed significantly higher *TUBB* mRNA compared to grade 2 in ERα-negative breast cancer patients (Figure 3D), whereas the differences between grades 3 and 1 and between grades 2 and 1 were not significant in ERα-negative breast cancer patients (Figure 3D).

### 3.4. Identification of the Genes That Positively and Negatively Correlate with TUBB in ERα-Positive and ERα-Negative Breast Cancer Patients

Taking into account that *TUBB* showed differential KM plots in ERα-positive and ERα-negative breast cancer patients, it was crucial to explore the genes that significantly correlate positively and negatively with *TUBB* in ERα-positive and ERα-negative breast cancer patients to explore the pathways in which TUBB is involved in, as described in the summary of the experimental design (Figure 4).

The genes that show significant positive correlation with *TUBB* and show a similar KM plot pattern (worse OS, RFS, and DMFS) in ERα-positive breast cancer patients were selected (Table 1).

Moreover, the genes that show significant negative correlation with *TUBB* and show opposite KM plot pattern (better OS, RFS, and DMFS) in ERα-positive breast cancer patients were also selected (Table 1). The same process was used to select the genes in ERα-negative breast cancer patients (Table 2).

### 3.5. Genes That Correlate with TUBB mRNA Expression in ERα-Positive and ERα-Negative Breast Cancer Patients Are Involved in Different Pathological Pathways

To identify the potential function of TUBB in ERα-positive and ERα-negative breast cancer, the genes that show significant positive correlation with *TUBB* in ERα-positive and ERα-negative breast cancer and that show similar KM plots pattern to *TUBB* were selected to identify the potential pathways that they are involved in. Additionally, the genes that show significant negative correlation with *TUBB* in ERα-positive and ERα-negative breast cancer and that show the opposite KM plots were also selected. In Figure 5A, genes that significantly correlate with *TUBB* mRNA expression in ERα-positive breast cancer patients were shown to be more involved in the TSC/mTOR pathway compared to the genes that correlate with *TUBB* mRNA expression in ERα-negative breast cancer patients (Figure 5B).

### 3.6. TUBB Correlates Positively with Several Gene Markers of Immune Cells in ERα-Positive Breast Cancer Patients and Negatively in ERα-Negative Breast Cancer Patients

Taken together, that fact that *TUBB* mRNA expression showed differential effects on OS, DMFS, and RFS in ERα-positive and ERα-negative breast cancer patients, and that the genes that correlate with *TUBB* in ERα-positive and ERα-negative breast cancer patients were shown to be involved in differential pathological pathways, prompted the exploration of the type of the immune cells infiltrated in ERα-positive and ERα-negative breast cancer patients. The results showed significant positive correlation between *TUBB* mRNA expression and gene markers of immune cells (CD79A, CD19, CD2, CD3E, and CD3D) in ERα-positive breast cancer patients (Figure 6, black bars), whereas negative correlations were observed between *TUBB* mRNA expression and several gene markers of immune cells (CD79A, CD19, CD2, CD3E, and CD3D) in ERα-negative breast cancer patients (Figure 6, grey bars).

### 3.7. Potential Clinical Targeting of TUBB

Approved drugs that directly target TUBB were identified using DRUGSURV (Table 3); three out of the six drugs that directly target TUBB are approved in breast cancer treatment including vinblastine, vincristine, and vinorelbine.

## 4. Discussion

In this research, the expression of *TUBB* in breast cancer was explored using two different platforms including GEPIA and bc-GenExMiner. The analysis showed that significantly higher *TUBB* mRNA expression in breast cancer patients was observed compared to normal breast tissue, suggesting that the expression of *TUBB* might predict the prognosis of breast cancer. A published report agrees with our finding, in which it was shown that several tubulin genes including *TUBB* were upregulated in breast-cancer tissues and tumour-adjacent tissues compared to normal breast tissue [10].We further explored the prognostic value of *TUBB* in ERα-positive and ERα-negative breast cancer patients using the KM Plotter. Given that ERα-positive breast cancer patients with increased *TUBB* showed worse prognosis (worse OS, RFS, and DMFS) and ERα-negative breast cancer patients with increased *TUBB* showed better prognosis (better OS, RFS, and DMFS), this suggests that the expression of *TUBB* might be a predictive biomarker for worse prognosis in ERα-positive breast cancer patients and better prognosis in ERα-negative breast cancer patients. Although the correlation between *TUBB* and prognosis was not studied extensively in breast cancer, several published report showed that *TUBB* mRNA expression significantly correlates with worse OS in lung adenocarcinoma, renal, and liver cancer patients [8,20,21].

Taking into account that *TUBB* mRNA expression correlates positively with lymph nodes examined positive in ERα-positive breast cancer patients and that *TUBB* mRNA expression correlates with worse DMSF in ERα-positive breast cancer patients, these observations suggest that *TUBB* might be implicated in the development of metastasis in ERα-positive breast cancer patients. It has been suggested that *TUBB* might be involved in mediating metastasis in lung adenocarcinoma, possibly through the interaction with miR-195 that targets *BIRC5* [8]; however the exact mechanism remains to be unclear.

The analysis revealed that *TUBB* might be involved in different pathways in ERα-positive and ERα-negative breast cancer, in which *TUBB* might be more involved in the activation of TSC/mTOR pathway in ERα-positive breast cancer as the genes that significantly correlate with *TUBB* in ERα-positive breast cancer were shown to be more involved in TSC/mTOR pathway compared to genes that correlate with *TUBB* in ERα-negative breast cancer. Several published reports indicated that this pathway plays an integral role in mediating metastasis, invasion, and proliferation in several cancers, including breast cancer [31,32].

Given that the mTOR pathway regulates the differentiation and function of T and B cells [33,34] and that significant positive correlations were observed between *TUBB* and several gene markers of B and T cells in ERα-positive breast cancer, these findings agree with our previous observation that *TUBB* might be more involved in the activation of the TSC/mTOR pathway in ERα-positive breast cancer compared to ERα-negative breast cancer. Although the role of *TUBB* in mediating immune cell infiltration has not been investigated extensively, several members of the tubulin family have been shown to play a role in immune cell infiltration including TUBA1C, which mediates immune infiltration in lung cancer [35]. In addition, TUBA1B has been shown to mediate the infiltration of several immune cells in hepatocellular carcinoma, including CD274 and CTLA4 [36]. Moreover, TUBB6 has been shown to correlate significantly with immune-infiltrating cells including CD8+ T cells in neck squamous cell carcinoma [37]. Taking into account that several members of the tubulin family have been shown to mediate the infiltration of immune cells, and that *TUBB* correlates positively with several gene markers of B and T cells in ERα-positive breast cancer, *TUBB* might be implicated in mediating the infiltration of immune cells in ERα-positive breast cancer.

Taking into account the fact that the analysis revealed that *TUBB* mRNA might be used as a predictive biomarker of worse prognosis in ERα-positive breast cancer patients and better prognosis in ERα-negative breast cancer patients, and that breast cancer treatments such as vinblastine, vincristine, and vinorelbine target TUBB directly, these results suggest that clinicians should avoid using these drugs in ERα-negative breast cancer patients and use them only on ERα-positive breast cancer patients.

## 5. Conclusions

In conclusion, the results indicate that *TUBB* could be used as a potential prognostic biomarker for worse prognosis in ERα-positive breast cancer and better prognosis in ERα-negative breast cancer, in which *TUBB* correlates with worse OS, DMFS, and RFS in ERα-positive breast cancer patients and better OS, DMFS, and RFS in ERα-negative breast cancer patients. The results suggest that clinicians should avoid using vinblastine, vincristine, and vinorelbine in ERα-negative breast cancer patients. In addition, the results showed that *TUBB* might mediate the development of metastasis in ERα-positive breast cancer patients possibly through the TSC/mTOR pathway. However, further experiments should be carried out to further explore the role of TUBB in ERα-positive and ERα-negative breast cancer.

## Figures and Tables

**Figure 1 diagnostics-12-02067-f001:**
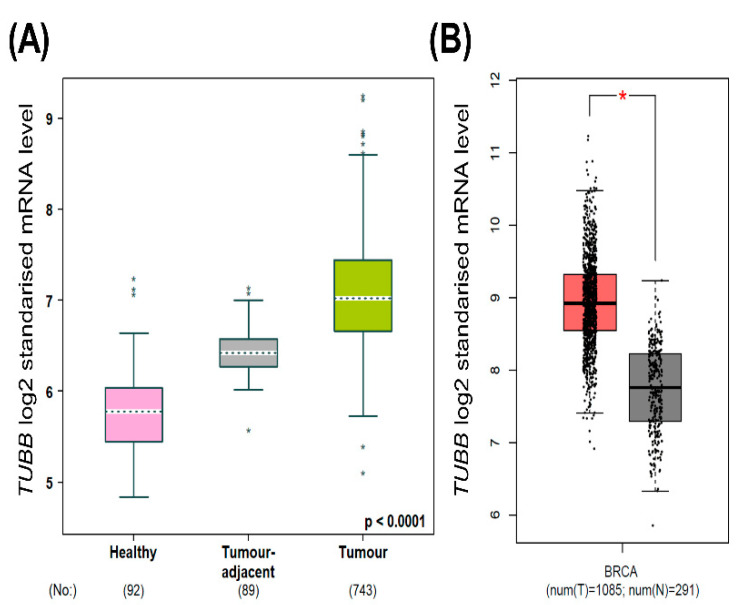
(**A**) Shows *TUBB* mRNA expression in normal, tumor-adjacent, and breast cancer tissues using bc-GenExMiner; (**B**) shows *TUBB* mRNA expression in normal and breast cancer tissue using GEPIA. * *p* < 0.05.

**Figure 2 diagnostics-12-02067-f002:**
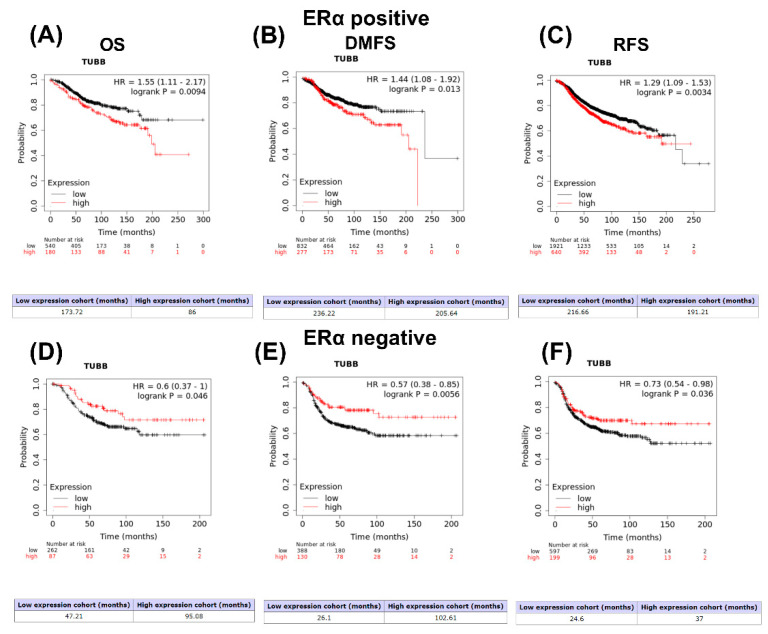
(**A**) Kaplan–Meier curve showing OS of *TUBB* in ERα–positive breast cancer patients; (**B**) Kaplan–Meier curve showing DMFS of *TUBB* in ERα–positive breast cancer patients; (**C**) Kaplan–Meier curve showing RFS of *TUBB* in ERα–positive breast cancer patients; (**D**) Kaplan–Meier curve showing OS of *TUBB* in ERα–negative breast cancer patients; (**E**) Kaplan–Meier curve showing DMFS of *TUBB* in ERα–negative breast cancer patients; (**F**) Kaplan–Meier curve showing RFS of *TUBB* in ERα–negative breast cancer patients.

**Figure 3 diagnostics-12-02067-f003:**
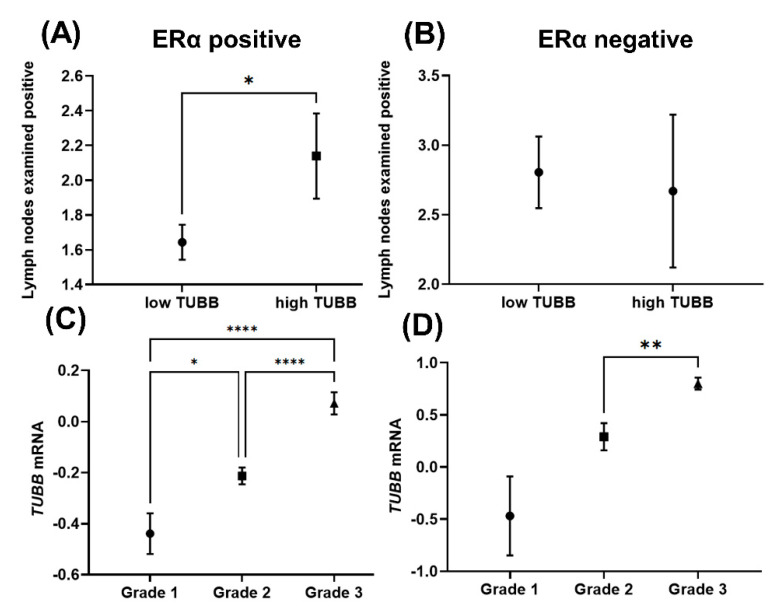
(**A**) Showing lymph nodes examined positive in high and low *TUBB* mRNA groups in ERα–positive breast cancer patients; (**B**) Showing lymph nodes examined positive in high and low *TUBB* mRNA groups in ERα–negative breast cancer patients; (**C**) Showing *TUBB* mRNA in neoplasm histologic grades in ERα–positive breast cancer patients; (**D**) Showing *TUBB* mRNA in neoplasm histologic grades in ERα–positive breast cancer patients. * *p* < 0.05, ** *p* < 0.01, and **** *p* < 0.0001.

**Figure 4 diagnostics-12-02067-f004:**
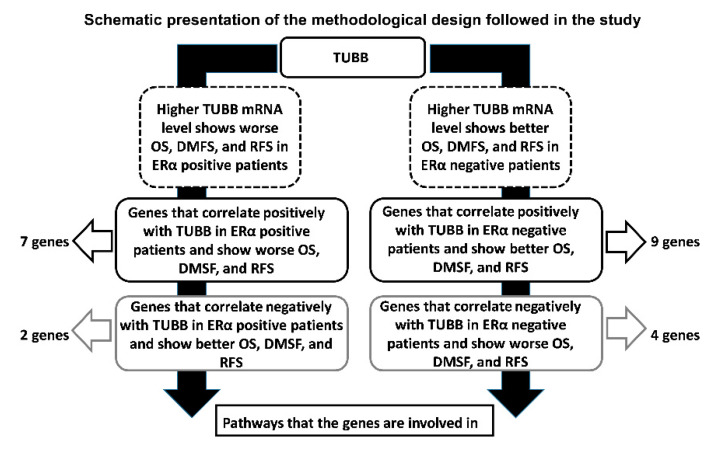
Shows schematic presentation of the experimental design followed in this study.

**Figure 5 diagnostics-12-02067-f005:**
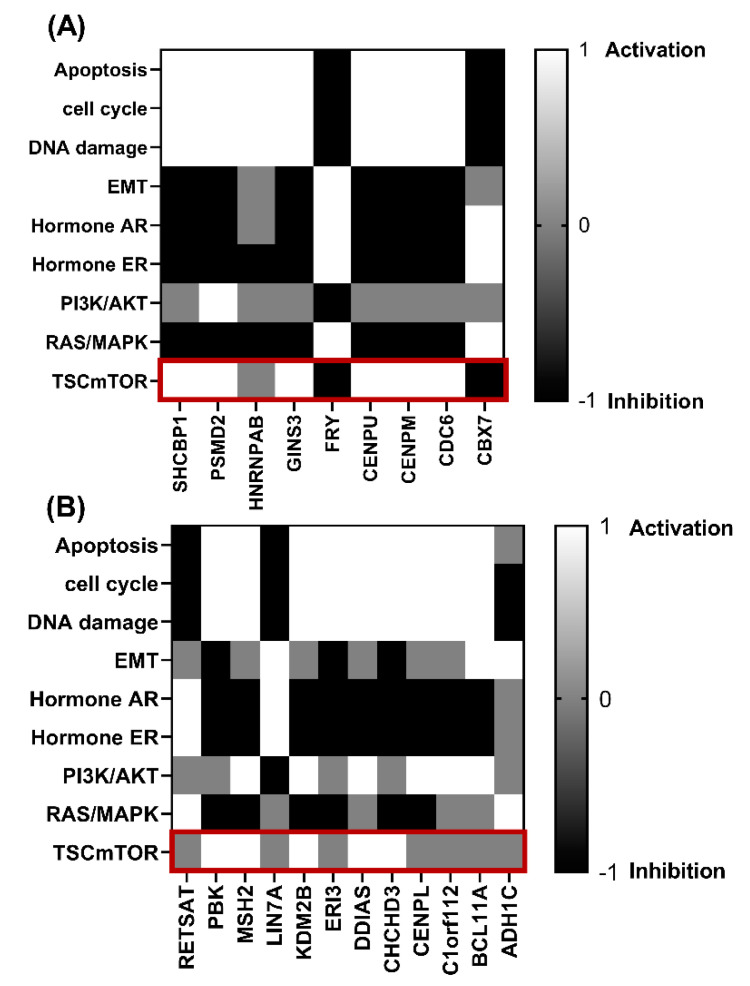
(**A**) Heat map shows the activated and inhibited pathways based on genes that significantly correlate with *TUBB* in ERα–positive breast cancer patients; (**B**) Heat map shows the activated and inhibited pathways based on genes that significantly correlate with *TUBB* in ERα–negative breast cancer patients.

**Figure 6 diagnostics-12-02067-f006:**
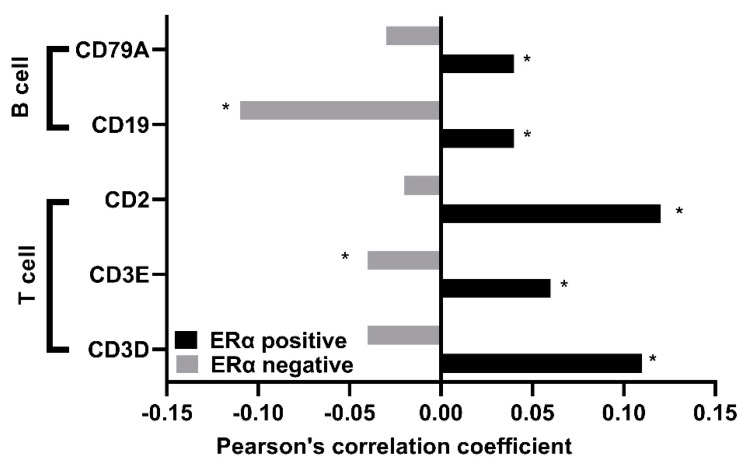
Shows Pearson’s correlation coefficient between *TUBB* mRNA and several gene markers of immune cells in ERα–positive and ERα–negative breast cancer patients. * *p* < 0.05.

**Table 1 diagnostics-12-02067-t001:** Shows genes that correlate positively and negatively with *TUBB* in ERα-positive breast cancer patients.

Gene Name	Correlation Coefficient ^1^	*p*-Value	Number of Patients
SHCBP1	0.4589	<0.0001	3685
GINS3	0.4468	<0.0001	3685
CDC6	0.4441	<0.0001	3685
CENPM	0.4228	<0.0001	3685
PSMD2	0.4178	<0.0001	3685
CENPU	0.4102	<0.0001	3685
HNRNPAB	0.4036	<0.0001	3685
CBX7	−0.4316	<0.0001	3685
FRY	−0.412	<0.0001	3685

^1^ Pearson’s correlation coefficient.

**Table 2 diagnostics-12-02067-t002:** Shows genes that correlate positively and negatively with *TUBB* in ERα-negative breast cancer patients.

Gene Name	Correlation Coefficient ^1^	*p*-Value	Number of Patients
MSH2	0.5704	<0.0001	510
CENPL	0.5002	<0.0001	510
PBK	0.4955	<0.0001	510
ERI3	0.4931	<0.0001	510
DDIAS	0.4771	<0.0001	510
KDM2B	0.4735	<0.0001	510
C1orf112	0.441	<0.0001	510
CHCHD3	0.4275	<0.0001	510
BCL11A	0.4164	<0.0001	510
LOC100505942	−0.432	<0.0001	241
RETSAT	−0.4088	<0.0001	510
LIN7A	−0.4068	<0.0001	510
ADH1C	−0.4006	<0.0001	510

^1^ Pearson’s correlation coefficient.

**Table 3 diagnostics-12-02067-t003:** Shows the drugs that directly target TUBB.

TUBB-Targeting Drugs ^1^	Chemical Formula	CID	Clinical Stage
Vincristine	C46H56N4O10	5978	metastatic breast cancer (phase II)
Vinorelbine	C45H54N4O8	5311497	approved for breast cancer
Vinblastine	C46H58N4O9	13342	metastatic breast cancer (phase II)

^1^ approved breast cancer drugs that directly target TUBB.

## Data Availability

Not applicable.

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
