# Peer review of "Bioinformatics Identification of TUBB as Potential Prognostic Biomarker for Worse Prognosis in ERα-Positive and Better Prognosis in ERα-Negative Breast Cancer"

_diagnostics, 2022, doi:10.3390/diagnostics12092067_

Round 1
Reviewer 1 Report
Comment to the authors:
This is an interesting study and the authors have predicted Tubulin β class I gene (TUBB) as a biomarker for breast cancer through different server-based analysis approaches. The paper is generally well written and structured but remains some grammar and English mistakes. However, in my opinion, the paper has some minor shortcomings regarding some data analyses and text, and I feel this point should be introduced before considering the manuscript in “Diagnostics”. Below I have provided some remarks regarding the manuscript.
1. The abstract of the manuscript should be improved.
2. I also suggested citing more relevant and recent literature in introduction section.
3. In material and methods, the subsection of each subheading should be provided for both material methods and results section.
4. In the section of “TUBB mRNA expression in normal breast tissue versus breast cancer tissue”, the number of samples used should be mentioned. During the expression analysis, why the threshold value set >1.5, please explain the reason.
5. The dataset used in this study should be provided in a table or excel file in supplementary file section.
6. The top 50 genes found during the analysis should also be provided in supplementary file section.
7. Genes that correlate significantly with TUBB in ERα-positive and negative breast cancer, upregulated or downregulated?. Please mentioned.
8. The drug that identified targeting “TUBB”, their chemical formula, structure, CID, and which stage of clinical development should be provided.
9. In several instances, I also suggested to do more relevant analysis like analysis of expression in clinical features and promoter methylation, and relationship between TUBB expression and survival of BC patients. The authors can take help from the following related papers:
I. https://doi.org/10.1016/j.heliyon.2020.e05087
II. https://doi.org/10.1007/s00109-021-02088-w
III. https://doi.org/10.1016/j.ygeno.2020.11.012
10. The conclusion and discussion section must be improved also the authors are highly recommended to support the analysis through “in-vitro” work.
Reviewer 2 Report
The interesting manuscript pointing to the potential role of TUBB as a biomarker in breast cancer. The authors point to the need for further research in ER negative cancers, which is also important information.
Author Response
N/A
Reviewer 3 Report
Overall the study ,which uses TUBB as biomarker is an interesting way forward to look at specific gene in detail in the prognosis of breast cancer. The findings are useful in further confirming the role of TUBB in ER+ and ER- . I have some suggestions to further improve the ms:
1) the statement "The analysis revealed that TUBB 20 might be considered as a predictive biomarker for worse prognosis in ERα-positive and better prognosis in ERα-negative breast cancer" is confusing. how do you assess worse and better.
2) the correlation between TUBB mRNA and several gene markers 205 of immune cells in ERα-positive and ERα-negative breast cancer patients were shown in results and it was not discussed clearly.
3) TUBB is a well studied and well researched gene. however not many references were cited. The references on the role of TUBB in cancer prognosis can be improved.
Round 2
Reviewer 1 Report
The author has substantially improved the manuscript and can be accepted for publications.
Author Response
N/A